# Enhanced intermolecular coulombic decay due to sulfur heteroatoms in thiophene dimer
Deepthy Maria Mootheril [1] ✉, Anna D. Skitnevskaya[2], Xueguang Ren[3], Mevlut Dogan[1],
Alexander B. Trofimov[2], Alexander I. Kuleff [4], Lorenz S. Cederbaum [4], Thomas Pfeifer [1] &
Alexander Dorn[1] ✉

Intermolecular Coulombic decay (ICD) is an important relaxation process of excited atoms and molecules in an environment, producing low-energy electrons that may contribute to radiation damage. Despite its significance, the mechanisms influencing ICD in molecular complexes remain unclear. Here, we investigate and unambiguously prove the ICD process in thiophene dimer, an aromatic ring with a third-row atom. Using multi-particle momentum coincidence spectroscopy, accompanied by high-level electronic structure calculations, we elucidate that the ICD process is initiated from the sulfur-containing inner-valence orbitals which are energetically below the Auger threshold. This leads to an enhancement in the emission of low-energy ICD electrons compared to other aromatic ring dimers. By utilizing this 'ICD-only decay' contribution we quantify ICD probabilities above the Auger threshold. This study reveals the pivotal role of sulfur in shaping the ICD electron spectrum, which can be implied to control the low-energy electron emission in biological systems.

High-energy primary radiation produces large number of secondary electrons along their tracks in matter. These low energetic electrons have typical energies ranging from zero up to several tens of eV and can cause further ionization in outer and also inner-valence orbitals of atomic or molecular systems. These collision processes and the subsequent molecular fragmentation have been studied regularly in the gas phase on smaller molecular units. More recently condensed phase studies came into the focus, where additional channels, involving neighboring molecules, are open. Such intermolecular reactions have been studied, e.g., in clusters of atoms or molecules where the constituents are weakly bound by hydrogen bonds or dispersion forces like the van-der-Waals interaction or $\pi$-$\pi$ interactions. One important reaction channel is interatomic/intermolecular Coulombic decay (ICD), which is a relaxation mechanism proposed by Cederbaum and coworkers[1] in which the atom/molecule deexcites via transferring a virtual photon to the neighboring atom/molecule and, thereby, ionizing it. Numerous experimental[2–5] and theoretical[6–9] studies of ICD in different, mostly atomic, systems were conducted which gave insights in the involved electronic states and in the decay dynamics (for recent review see[10]).

ICD induced by electron collisions was studied first in argon dimers in the gas phase using a reaction microscope (ReMi) in which all five outgoing charged fragments were detected in coincidence[11]. The ensuing discussions concerned the implications of ICD on biological systems, which could, e.g., lead to DNA strand breaks upon irradiation by low-energy electrons[12]. In an attempt to better understand the underlying processes, ICD was studied in hydrated DNA building blocks like tetrahydrofuran, a surrogate molecule of sugar deoxyribose in the DNA structure. It was found that ICD can be initiated by ionizing an inner-valence orbital of oxygen in the hydrogen-bonded water molecule[13]. Furthermore, ICD was also identified in pure carbon-based aromatic systems, such as benzene dimers in which the constituents were bound by $\pi$-$\pi$ interactions and CH-$\pi$ interactions[14]. It was proposed that the kinetic energy release (KER) spectrum from the Coulomb explosion (CE) following fast ICD can serve to identify the initial inter-molecular distances and, thus, to determine the structures of such organic dimers[15]. Based on these principles, experiments have shown that the benzene dimer has a predominant $T$-shaped structure[14]. More studies exist for dimers of heteroaromatic molecules like pyridine, pyrimidine etc., with nitrogen as the heteroatom[16]. While the $T$-shaped geometry with CH-$\pi$ interactions prevails in benzene, $\pi$-$\pi$ interactions begin to compete and even dominate, when one of the benzene rings is replaced by a nitrogen-containing six-membered heterocycle. Furthermore, if the number of

¹Max-Planck-Institut für Kernphysik, Saupfercheckweg 1, Heidelberg, Germany. ²Federal Research Center, A. E. Favorsky Irkutsk Institute of Chemistry, Siberian Branch of the Russian Academy of Sciences, Favorskogo str. 1, Irkutsk, Russia. ³MOE Key Laboratory for Nonequilibrium Synthesis and Modulation of Condensed Matter, School of Physics, Xi'an Jiaotong University, Xi'an, China. ⁴Theoretische Chemie, Physikalisch-Chemisches Institut, Universität Heidelberg, Heidelberg, Germany. ✉e-mail: deepthy.mootheril@mpi-hd.mpg.de; dornalex@mpi-hd.mpg.de

nitrogens in the system increases, hydrogen bonding begins to outcompete the other types of interactions[16].

In the present work, we study ICD in dimers of thiophene, which is the basic sulfur-containing five-membered heteroaromatic molecule. It is of great importance for many fields of contemporary chemistry, biochemistry, and technology[17]. Several studies focused on the ionization or excitation from the outer and inner-valence orbitals of thiophene are published[18–21]. Photoelectron spectroscopic results obtained using different light sources covering 584 and 304 Å helium emission lines and tunable synchrotron radiation gave detailed insight in the valence orbital binding energies and the satellite states of thiophene monomers[19,22–24]. In addition, several electron momentum spectroscopic measurements have been carried out using (e,2e) techniques, which enabled the study of electron momentum distributions of the valence orbitals[25,26]. Electron collision experiments were also conducted to measure total elastic scattering and ionization cross-sections over a broad range of impact energies, which benchmarked the available theoretical calculations[27–29]. The main consequence of the presence of the sulfur heteroatom is the increase in polarizability of the molecule[30] and the increase in the $\pi$-electron density in comparison to homocyclic hydrocarbons like benzene. This in turn increases the non-covalent interactions between the constituents in dimers and clusters[17]. On the other hand, the permanent dipole moment of thiophene is significantly smaller than for aromatic rings with the heteroatoms nitrogen or oxygen. Thus, several conformational geometries were theoretically predicted for thiophene dimers and larger clusters in comparison to benzene, which shows mainly four different conformational geometries[31,32]. Such increased non-covalent interactions can modify the valence-orbital spectrum and, in case of thiophene, lead to an enhancement in the non-local decay mechanisms, such as ICD, upon ionization from the inner-valence vacancies.

Here, we explore the ICD process in thiophene dimers after inner-valence ionization upon electron collisions of energy 68 eV and we study the influence of the sulfur heteroatom on the ICD dynamics. We employ coincident detection of several charged particles (one electron and two ions) arising from the reaction. A similar coincidence measurement on the thiophene dimer exists, detecting ion pairs undergoing CE and conducting an in-depth investigation of the dimer's structure in a supersonic gas jet[33]. This study also determines the potential energy curve of thiophene dimers in different conformations using the measured KER spectrum of the ion pairs. However, coincident electron detection was not performed, which is crucial for confirming the ICD process and measuring ICD electron energy.

## Results

In our experiment, we study the following CE channel of thiophene dimers.

$$e^- + (C_4H_4S)_2 \rightarrow (C_4H_4S)_2^{+*} + 2e^- \quad (1)$$

$$\rightarrow 2\,C_4H_4S^+ + 2e^- + e^-_{ICD} \quad (2)$$

The process is depicted in the schematic in Fig. 1. Upon impact of electrons with 68 eV energy, the dimer is ionized in the inner-valence shell leaving it in an excited ionic state. Upon relaxation, an outer-valence electron fills the inner-valence vacancy and the excitation energy is transferred via Coulomb interaction to the neighbor, which is ionized. This ICD process is ultra-fast with a lifetime in the ten to hundred femtosecond range[7]. Therefore, ICD enables to dissipate the internal electronic excitation energy faster than other relaxation processes can do. For isolated multi-atomic molecules the main relaxation process of inner-valence vacancies is internal conversion (IC), where the electronic energy is converted into molecular vibrations that eventually lead to the fragmentation of the ion. In environment, the molecule can also relax by ICD populating directly low-lying cationic states that not necessarily lead to a fragmentation. ICD thus shows different signatures, which we will identify in the experimental data. Firstly, there is the subsequent CE with the two parent ions gaining equal momenta in the opposite directions resulting in their back-to-back emission. If ICD is fast compared with any intermolecular motion, the observed KER should be

consistent with the initial separation of the molecules in the neutral dimer. Secondly, the minimum energy loss of the scattered projectile required to produce the ion pair identifies the initial inner-valence vacancy with the least binding energy which can initiate ICD. Thirdly, there is a characteristic low-energy electron, called ICD electron, that is ejected in the ICD process.

In order to identify these signatures experimentally, two ions and one of the outgoing electrons are detected in coincidence.

### Geometry determination of thiophene dimers from the KER spectrum of CE events

The thiophene ions from the CE of the dimer are emitted back-to-back and, therefore, are located on the diagonal in the ion-ion momentum coincidence map. This is shown in Fig. 2c for the $x$-components of the momentum vectors which is the direction perpendicular to the incoming electron beam and the supersonic gas jet. The ion-ion time correlation map is shown in Fig. 2a. Ion pairs from a two-body CE are lying on the weakly visible diagonal line inside the red dashed rectangle. There is a strong background of events besides the diagonal, where neutral particles are emitted in addition to the two ions. They originate from hydrogen loss channels, where one or both of the dimer constituents lose one or more hydrogen atoms due to vibrational excitations after ionization[34]. In addition, CE events of larger clusters leading to the production of two thiophene ions and neutrals also appear in this region of the time coincidence map. In both cases, the momentum from the Coulomb repulsion is shared between two ions and the outgoing undetected neutrals. Therefore, the ions do not give rise to a sharp diagonal line, but instead a blurred distribution around the diagonal.

To filter the desired CE events of the dimer from the background events we use two techniques: (1) Selecting events where the sum of the ions' times-of-flight (TOF) falls within a 100 ns window, reducing false coincidences and some diagonal background. (2) Exploiting Coulomb repulsion, where ions are emitted back-to-back with nearly zero total momentum. Considering the momentum spread due to the limited resolution, the events arising from the CE of dimers leading to ($C_4H_4S+$, $C_4H_4S+$) are separated from the background imposing the condition of small sum momentum (<20 a.u.) (Fig. 2b) of the detected ions in addition to a non-zero KER. Further details, including momentum coincidence maps, are provided in SM, Section I. Finally, the KER from the CE is calculated as the sum of the kinetic energies of the individual fragment ions.

Figure 2d shows the resulting KER distribution from the CE of dimers. There is a peak at 2.6 eV and small tails around 1.8 and 3.4 eV. The KER

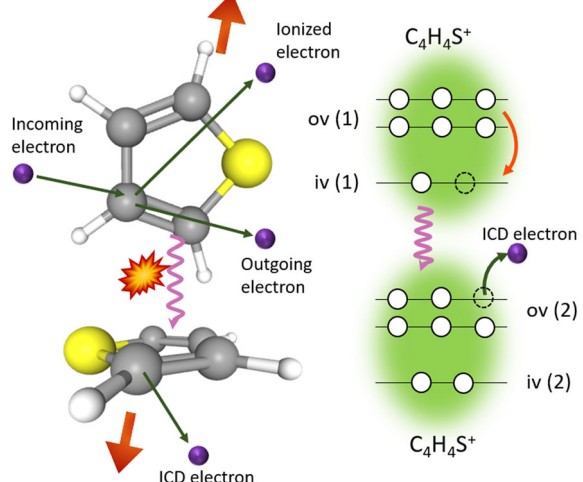

**Fig. 1 | Schematic of ICD process in thiophene dimers[64].** The inner-valence vacancy created by the incoming electrons of 68 eV is filled by the outer-valence electron (orange arrow on the right hand side) and the relaxation energy is transferred to the neighbor. If it is energetically allowed, ionization takes place on the neighboring molecule and a Coulomb explosion (CE) follows as indicated by the red arrows on the left-hand side.

**Fig. 2 | Time and momentum coincidence map of two fragment cations from the thiophene dimer and the corresponding KER spectra. a** The time-of-flight coincidence map which includes the ICD events (red dashed rectangle) and the background events. For details about the background refer text. **b** The sum of the momenta of the ions versus the KER. The small island within the red dashed circle shows the CE events from the dimers. **c** The momentum coincidence map of ions along the $x$ direction from the CE events of thiophene dimers, separated from the background. **d** The measured KER spectra for CE of thiophene dimers. The red bars show the AIMD simulation for the $T$-shaped conformer[33], i.e. the energetically most stable structure.(I)[31].

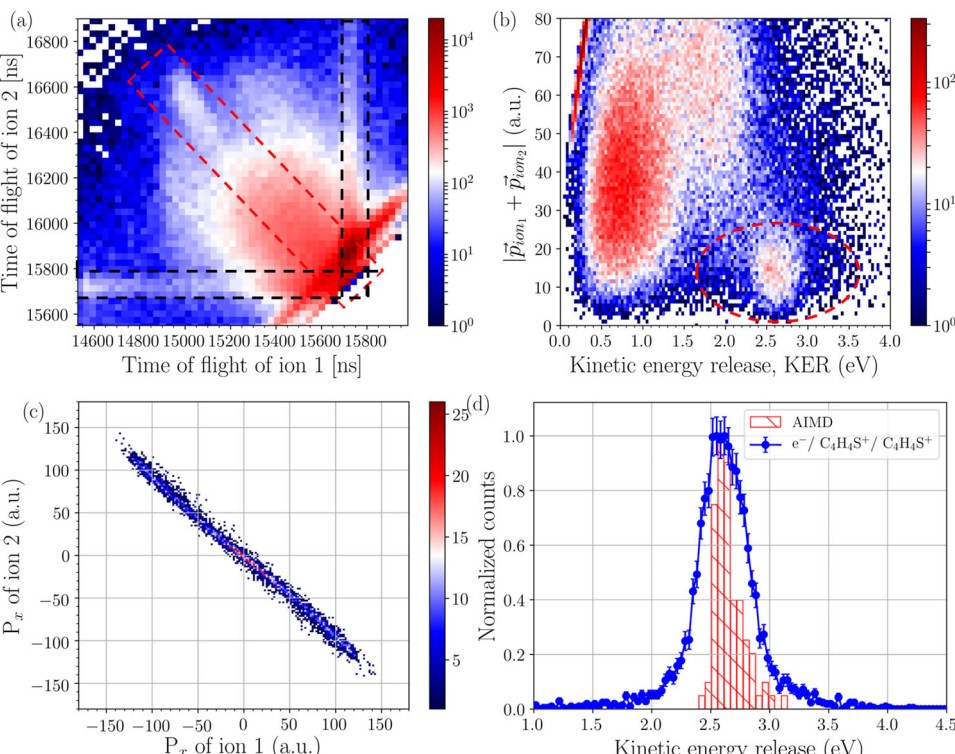

spectrum is analogous to Fig. 4 in[33]. The thiophene dimers exist in several conformational geometries in the gas phase. Previous calculations show that out of 17 conformers suggested in the literature[31] a $T$-shaped (I) conformer is the most stable structure. The measured KER distribution is compared with that obtained by ab initio molecular dynamics (AIMD) simulations of the CE of the dimer with $T$-shaped (I) conformational structure. The detailed description of the simulation and the structural determination can be found elsewhere[33]. The observed position of the maximum of the distribution (~2.6 eV) is at lower energy than the Coulomb-potential energy for the equilibrium inter-molecular separation (5 Å)[31] of the $T$-shaped conformer which is around 2.9 eV. This is a general trend in the CE of molecules[14,34] where part of the Coulomb energy is transferred to vibrational and rotational degrees of freedom. The agreement between the maxima of the experimental data and the AIMD simulations shows that a large fraction of the dimers in the supersonic gas jet have a T-shaped conformational structure. The presence of the tails on both sides of the main peak could be due to a smaller fraction of other prominent conformers, namely conformer B and K[31], which are expected to produce a KER at 2.2 eV (conformer $K$—coplanar structure) and at 3.6 eV (conformer B—sandwich structure)[33], respectively. The presence of other conformational structures cannot be completely ruled out.

### Identification of the inner-valence vacancy initiating the ICD process

The energy spectrum of electrons measured in coincidence with a thiophene ion pair ranges from very small energies close to zero up to maximum of $E_0-\Delta E$. Here, $E_0$ is the incoming projectile energy and $\Delta E$ is the minimum energy required to produce the ion pair. Typically, the electrons with low energies are the secondary electrons from the ionizing collision and from the ICD process and the faster electrons are the scattered projectiles. Therefore, we can analyze these distinct energy regions separately and obtain information on the individual energy spectra of the three active electrons in the ICD process. In Fig. 3a, projectile energy-loss spectra are shown, which are measured in coincidence with different ion species. The energy loss is $E_0-E_1$, where $E_1$ is the energy of the scattered projectile. The onset energy corresponds to the lowest energy to produce an electron vacancy from which the detected ion is produced. The diagram shows spectra for the CE channel of the dimer and for two dissociation channels of the isolated monomer, which

lead to $C_3H^+$ and $C_2H_2^+$ ions (black open, violet solid, and orange solid circles, respectively). For these three channels, the spectra are very similar, in particular for the onset which indicates that the ionization takes place from the same inner-valence orbitals.

It has been reported previously that the appearance energies of these fragment ions are $21.4 \pm 0.2$ eV $(C_3H^+)$ and $19.0 \pm 0.8$ eV $(C_2H_2^+)$[18]. Therefore, we conclude that the same inner-valence vacancies which lead to these fragments in monomers lead to ICD in dimers. Concerning the positions and widths of the onsets in Fig. 3a, we compare with the simultaneously recorded energy-loss spectrum for ionization of helium atoms with a binding energy of 24.6 eV, which is also shown. For atoms the physical onset is a step-function at the ionization energy, which is broadened by the instrumental energy resolution. This is mainly due to the spectrometer energy resolution for a scattered projectile with around 43 eV, which is about 4 eV, while the energy width of the primary electron beam is comparably small (<0.5 eV). For the molecular ionization also the physical onset can be broadened, e.g. due to the influence of Franck-Condon factors in the transitions to a range of vibrational states in the molecular ion[35]. Additionally, with increasing energy loss more ionic states can be reached and contribute. Accordingly, the onsets for the molecular ion channels shown are in the range of 20–22 eV and are broader than that for helium.

The bottom panel in Fig. 4 shows the calculated single ionization (IP) spectrum of thiophene dimers with T-shaped geometry computed at the ADC(3)/cc-pVDZ level of theory (black sticks and convolution line, see computational methods for details). There is an intense group of states in the range of 20–23 eV, which is produced from the ionization of C 2s and S 3s inner-valence orbitals corresponding to $7a_1^{-1}$ and $4b_2^{-1}$ states of thiophene molecules, the average contribution of sulfur to the band is about 30%. The position of this band is within the experimental accuracy consistent with the measured onset of the electron energy loss spectrum.

### Measured ICD electron energy spectrum

The experimental ICD electron spectrum is obtained from the measured data by comparing the low-energy electron spectra for the ionization channels discussed above, the monomer dissociation (coincidence with $C_3H^+$ or $C_2H_2^+$) and the dimer CE channels. Both are shown in Fig. 3b. For monomers the detected electron originates from the ionizing collision only,

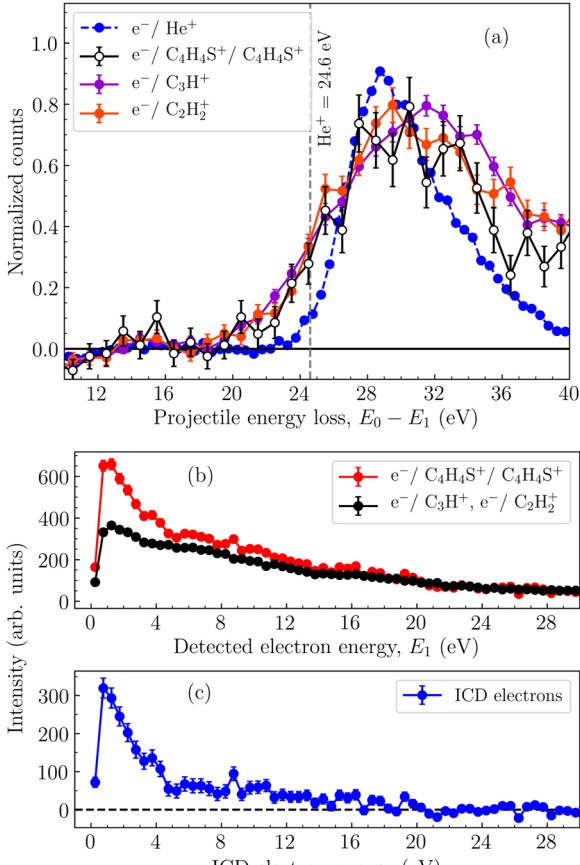

**Fig. 3 | Measured energy spectra for scattered projectile and ejected electrons.**
**a** Energy-loss spectra for CE channel of thiophene dimers (black open circles),
monomer inner-valence ionization leading to $C_3H^+$ (violet solid circle),
$C_2H_2^+$ (orange solid circle) and helium single ionization (blue dashed line). The gray
dashed vertical line indicates the binding energy of helium at 24.6 eV. **b** The low
energy electron spectra for the CE channel and monomer inner-valence ionization
channel. The curves are normalized from 25 eV to higher energies, since below 25 eV
we expect enhancement due to ICD electrons. **c** The difference between both curves
in Fig 3b gives the ICD electron spectrum (for the explanation see the text).

while for dimers it can originate from both, the ionizing collision or the ICD
process. For both target species, the ionized inner-valence orbitals are the
same and, therefore, also the electron energy distributions for the ionizing
collision are expected to be the same. To extract the ICD electron energy
distribution from the dimer spectrum, both spectra are normalized to the
integral of the counts above 25 eV where no ICD electrons are expected.
Finally, the difference between the normalized spectra is obtained and
provides the pure ICD electron energy distribution (blue solid circles in
Fig. 3c). It shows the highest intensity below 5 eV and extends up to about
20 eV. Below 0.7 eV there is zero electron detection efficiency due to the
projectile beam dump in the center of the electron detector.

**Theoretical single and double ionization spectrum for *T*-shaped
thiophene dimer**
In the following discussion about the theoretical calculations, we present
only the results for the *T*-shape dimer, since it is dominating the target
sample according to our KER spectrum. We note, however, that while our
modeling predicts similar behavior also for the sandwich dimers, compar-
ison can be found in the SM, section III.

As described above, both local and non-local electronic decays of singly
ionized states end up with doubly charged systems, thus theoretical mod-
eling of the double ionization spectrum (DIP) spectrum provides access to
all possible final states and identification of the energetic thresholds beyond

which the decays can occur. Here, the double ionization spectrum of the
thiophene dimer is calculated at the ADC(2)/cc-pVDZ level of theory and
shown in the middle panel of Fig. 4. The states are separated into two groups
according to the localization of the vacancies. The final 'delocalized' states,
which can be reached via ICD have one charge on each of the dimer
molecules (middle panel, blue sticks), while 'localized' Auger states have
both charges on one of the molecules (middle panel, orange sticks). The
latter final states with localized charges can also be reached by the process
called electron transfer mediated decay (ETMD)[36,37]. In this process, the
inner-valence vacancy on one molecule is filled by an outer-valence electron
of the neighbor and the energy released is used to ionize the neighbor itself.
However, the rate for the ETMD process is much smaller than Auger decay
because it is only efficient at small intermolecular distances[36]. According to
our theoretical data, the lowest energy dicationic state created by the ICD
process is at 20.3 eV and the threshold of the Auger/ETMD process is at
23.3 eV. Thus, singly ionized states in the 20.3–23.3 eV energy range can
only undergo ICD decay (the energy gap highlighted by the green rectangle
in Fig. 4), while for higher lying states both decay channels are open with an
unknown branching ratio. However, Auger decay is generally considered to
be around one order of magnitude faster than the ICD[38].

**ICD electron energy spectrum and ICD efficiency**
From the theoretical IP and DIP spectra, an estimation of the ICD electron
spectrum can be gained, which provides an opportunity to compare with the
experimental result. The calculated ICD electron energy spectrum is shown
as the blue and green envelope in the top panel of Fig. 4 and the experimental
spectrum is shown in a solid red line for comparison. To that end, the ICD
electron kinetic energy distribution was derived as the energy difference of
each decaying state from the IP spectrum (bottom panel in Fig. 4) with all
possible ICD final states of lower energy (blue sticks in the middle panel of
Fig. 4). The corresponding spectral intensities were obtained as the product
of the spectral intensities of the inner-valence states and those of the doubly
ionized product states. This approximation assumes that the probability of
the transitions is independent of the energy differences between the initial
and final states and independent of the exact positions of the two separated
charges in the final state. To account for the different ICD probabilities from
states with closed and competing Auger/ETMD decay channels, a factor of
0.3 was applied to the intensities of the electron spectrum obtained from the
IP states above the Auger/ETMD threshold. This factor determines the
relative contributions of the blue and green contributions in the top panel of
Fig. 4 and is chosen to provide the best visual fit between experimental and
theoretical results along the high-energy tail. Such scaling qualitatively
accounts for the decrease in ICD decay probabilities due to the presence of
other decay channels. This 'competing' part of the ICD electron spectrum is
shown in blue and has a broad energy distribution due to the wide energy
range of the initial IP states formed roughly up to 35 eV. The part of the
electron spectrum (denoted as 'direct decay') produced by the decay of the
IP states where only ICD is possible (highlighted by the green rectangle in
the lower panel) is added to the competing contribution and is shown as the
green envelope in the ICD electron spectrum. The good agreement between
theory and experiment obtained in that way shows that the enhancement in
the low-energy ICD-electron intensity is due to the 'direct decay' con-
tribution where the ICD does not compete with other decay processes.
Whereas the 'competing' part of the ICD spectrum shows that ICD is not a
minor channel in the thiophene dimer even when other competing decays
are open. We note that the 30% probability used in the fit for the ICD is
higher than usual when also Auger/ETMD are opened[9,39,40] showing that
ICD is especially efficient in thiophene dimer.

**Discussion**
The key insights from the present study of the thiophene dimer are twofold.
First, the lowest energy inner-valence vacancy that initiates the ICD process
(identified from the onset of the projectile energy loss spectrum) is observed
at the calculated ICD threshold which is well below the DIP threshold of the
monomer[41] and the calculated Auger/ETMD threshold of the dimer

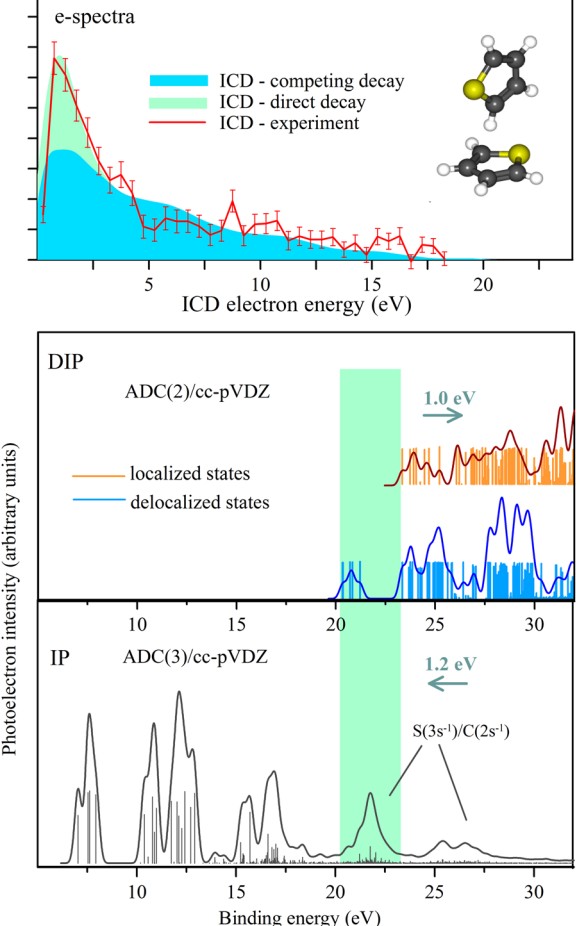

**Fig. 4 | Theoretical spectra of a *T*-shaped thiophene dimer.** Bottom panel - IP. Middle panel - DIP. States of different types are plotted separately, the delocalized states (final for ICD) are plotted in blue and the states localized on one thiophene molecule are plotted in orange. The scaled-down convolutions for both groups of states are plotted over for better visualization. The green rectangle shows the energy region between the lowest delocalized and lowest localized states (20.3–23.3 eV), where singly ionized states can only decay via ICD. Upper panel - simulated ICD electron spectrum, where the blue coloring indicates the ICD electrons coming from the IP states above 23.3 eV, the intensity is divided by 3 (see text for details) and the green coloring indicates the additional intensity of ICD electrons coming from the 20.3–23.3 eV IP states distributed on top of the contribution from the competing region. The red curve is the experimental data.

(middle panel in Fig. 4). This observation clearly indicates that the detected CE events result from ICD, providing unambiguous proof that ICD occurs, as it is the only significant decay mechanism available below the Auger threshold, which can lead to the CE of two thiophene ions. This is closely related to the comparatively large energy gap (~3.5 eV) between the dicationic states with charges on both sites and charges on the same site of the dimer, which can be seen in the calculated DIP spectra and is also predicted for other aromatic dimers (Fig. 5). It has previously been shown that the energies of secondary electronic decays can be significantly altered as a result of intermolecular donor–acceptor interactions[42]. Likewise, the observed considerable gaps are likely due to the possibility for redistribution of the vacancies over the aromatic ring.

Second, compared to previously studied aromatic ring systems a particularly strong enhancement of the ICD electron intensity is observed in the low-energy region below 4 eV in Fig. 3b. If we take the peak magnitude of the dimer spectrum (red full circles) relative to the monomer spectrum (black full circles) we obtain ~78% enhancement. This is compared to aromatic ring systems, such as benzene dimers with ~46%[14], pyridine dimers with

~27%[16], and hydrated tetrahydrofuran with ~22%. From the above discussion, and based on the very good agreement between the experimental and theoretical results, we can attribute this strong enhancement of the low-energy ICD electrons to the decay of the intense band of the inner-valence IP region, which has a significant contribution from the sulfur. Figure 5 shows theoretical IP spectra and the thresholds for local and nonlocal electronic decay processes for various aromatic dimers-furan, pyrrole, benzene, pyridine, and imidazole-along with the double ionization threshold of their respective monomers. The dimer geometries are sourced from[43,44], and an energy red shift of 1.2 eV is applied to all the IP spectra, justified by monomer spectrum comparisons[20,45,46]. Similarly, an energy blue shift of 1 eV is applied to the indicated double ionization thresholds of both dimers and monomers. One can see that in the case of thiophene, unlike the other dimers, the majority of the inner-valence states that are open for electronic decays appear within the gap between the thresholds of non-local and local dicationic states, allowing decay exclusively via ICD and ensuring the observed enhancement of low-energy ICD electrons. The only exception is benzene where the number and density of states between the ICD and Auger thresholds are comparable to those in thiophene. However, experiments show that the lowest in energy inner-valence state that decays via ICD in benzene is around 25 eV[14], which is very close to the double-ionization threshold of the monomer. A similar experimental observation was also found in pyridine[16]. However, for pyridine, the majority of states populated by removing inner-valence electrons are beyond the Auger threshold (Fig. 5), because the states contributed by nitrogen lie at higher energies beyond 25 eV. This suggests that in benzene and pyridine, ICD occurs only in competition with Auger decay. These experimental results in benzene/pyridine imply that either the states appearing between the ICD and Auger thresholds are not efficiently populated by an electron impact or there are other competing processes, like ultrafast non-adiabatic relaxation[47–49], that can quench the ICD from this part of the spectrum. It is particularly noteworthy that sulfur being a third-row period element contributes to the inner-valence region between 20–28 eV. In thiophene, however, due to the presence of sulfur, the most pronounced band appears between 20–23.3 eV which can decay only by ICD. We can, therefore, conclude that in thiophene this band is very efficiently populated and the follow-up ICD process has no competitors, resulting in the observed enhancement of the electrons emitted in the low-energy region. Our current findings of ICD makes thiophene dimers the first aromatic ring system to exhibit 'direct ICD decay' at energies below the Auger threshold.

Generally, the IP and DIP spectra of dimers are different from those of the isolated monomers due to intermolecular interactions like additional polarization of the neighboring molecule[15,23], which can even depend on the dimer conformer geometry[15]. As can be seen from the comparison presented in the SM, section III the IP spectra of the thiophene monomer and the *T*-shaped dimer are quite similar, while for the latter the entire spectrum is red-shifted due to intermolecular interactions. To investigate the differences arising from the mutual orientation of the thiophene molecules in the dimer, we calculate the IPs and DIPs spectra for the sandwich conformer (B), which can also contribute to the experimental sample. The results can be found in the SM, section III. Compared to the *T*-shape geometry, there are only minor differences in the IP and DIP spectra, e.g., for the *T*-dimer the bands of the IP spectrum are slightly broader, which can be an indication of a more pronounced donor–acceptor nature of the intermolecular interactions. Therefore, it can be concluded that for the system considered, the shape of the ICD electron spectrum depends only weakly on the orientation of the dimers and the influence of other conformational geometries, if present in the sample, can be neglected.

Finally, we discuss possible other pathways leading to the doubly charged dimer with charges on both centers eventually undergoing a CE. One such pathway is radiative charge transfer (RCT) from a doubly ionized state of one of the centers. In that process, an electron is transferred at an intermolecular separation lower than the equilibrium intermolecular separation leading to the formation of charges on both centers[11]. In such a case, the KER distribution should be at a larger value than the KER value

**Fig. 5 | The ADC(3)/cc-pVDZ calculations of inner-valence IP spectra of several heterocyclic dimers.** Panels (**a**)–(**f**) show the calculated inner-valence single ionization spectra of thiophene, furan, pyrrole, benzene, pyridine and imidazole dimers, respectively. The energy thresholds of ICD, Auger, and monomer double ionization are indicated as blue, orange, and green dashed lines, respectively. The thresholds are evaluated at the level of ADC(2)/cc-pVDZ level of theory.

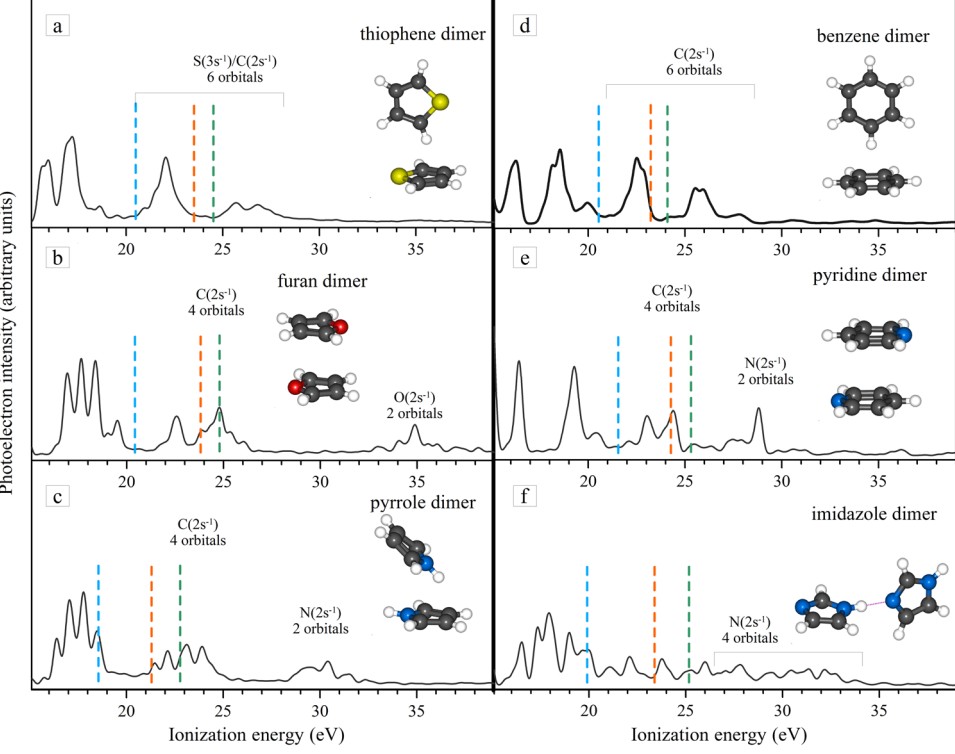

expected from the equilibrium intermolecular separation. Since, we have a weak tail in the KER spectrum at large energy, one cannot rule out such a pathway. However, since the onset of the CE channel is around 20 eV, which is lower than the double ionization threshold for localized states (23.3 eV) we can ignore the possibility for RCT because RCT requires an initial state higher than the double ionization threshold. Additionally, RCT is much slower than ICD[3,50]. Therefore, the ionization of the inner-valence orbital below the double ionization threshold in dimers or larger clusters should preferably proceed through ICD, if the other competing channels like fluorescence decay or IC are comparatively much slower.

The remaining possibility to initiate a CE is a sequential double ionization (SI) process where the incoming projectile ionizes the two monomers from the outer-valence orbital sequentially. The minimum energy required for the ICD and the SI processes are the same, namely two times the lowest ionization potential energy plus the Coulomb repulsion between the two ions. Therefore, a fraction of the CE signal might be due to SI. An estimate of the fraction of SI events in the total CE signal can be obtained by comparing the partial ionization cross-section for the ICD channel and SI channels. It was estimated that the SI constitutes 13.4% of the total CE events (for details see, SM, section IV). Thus, it can be concluded that SI is a minor channel in comparison to ICD.

## Conclusions

In summary, we have presented the investigation of the ICD process in thiophene dimers by studying their CE using the multi-particle momentum coincidence spectroscopy technique. By comparing the KER distribution of the two detected fragment ions undergoing CE events and the ab-initio molecular-dynamics simulation, we conclude that the majority of the ionized thiophene dimers have *T*-shaped geometry. From the onset of the electron energy loss spectrum of the electron detected in coincidence with the ions and the theoretical single-ionization spectrum of the thiophene dimer in a *T*-shaped configuration we unambiguously proves the ICD process that is initiated from the sulfur-containing $7a_1$ and $4b_2$ bands of inner-valence orbitals. Furthermore, the ICD electron energy spectrum is obtained from the electron energy spectra of the CE channel and the

monomer ionization channels and can be excellently understood using our theoretical results.

Notably, a particularly large energy gap observed between the lowest final dicationic state of electronic decays with shared charges and with both charges on one molecule of thiophene in the dimer suggests that ICD can be initiated from inner-valence states around 3.5 eV lower in energy than the Auger/ETMD decay threshold and can decay only via ICD. This leads to strong enhancement in the low-energy region in comparison with the previously measured dimers of single-ring aromatic molecules, including benzene[14], pyridine[16] etc, where this 'direct ICD-only decay' contribution is not observed. Based on our theoretical results we analyze this peculiar enhancement and it was found to be due to the decay of the intense band of inner-valence vacancies located in the energy range 20–23.3 eV, which has a significant contribution from sulfur. It should be noted that inner-valence bands originating from the common second-row period heteroatoms *N* and *O* are usually found deeper in the energy where local and non-local decay processes compete. Therefore, one can attribute the ICD enhancement observed here to the role of sulfur heteroatom. On the other hand, the calculated contribution to the ICD electron energy spectrum from the inner-valence bands above the Auger/ETMD threshold was multiplied by a factor of 0.3 for the best visual fit with the experimental spectrum. This factor approximately quantifies the reduction in ICD decay probability when other decay channels are present and shows that ICD is a potent process, in agreement with what was also shown for the benzene dimer[14].

## Method
### Experimental methods
The experiment was done using a ReMi[51] that is specially designed for electron impact ionization studies. The detailed description of the apparatus can be found elsewhere[11]. The dimers of thiophene were prepared using a supersonic gas jet. Helium gas is used as a carrier gas and is bubbled through a reservoir containing liquid thiophene with a vapor pressure of approximately 106 mbar at room temperature[52]. Thiophene vapor coexpanded with the helium carrier gas at 1 bar through the nozzle of 30 μm diameter, where clusters are formed by three-body collisions with the carrier gas. The gas jet

containing the thiophene dimers is ionized by an electron beam of 68 eV energy with a pulse duration of 0.5 ns. The electron pulses are produced from a photoemission electron gun, which is focused to a diameter of ~1 mm. From the mass-discriminated cluster ion yields, an approximate dimer-to-monomer ratio of 2% and trimer-to-monomer ratio of 0.14% are estimated. Using homogeneous electric and magnetic fields the charged fragments are extracted from the interaction region. Electrons are extracted using a 2 V/cm electric field and a 6.9 G magnetic field whereas for the extraction of heavy ionic fragments, the magnetic field is maintained and the electric field is ramped up to 23 V/cm around 200 ns after the electrons are detected. The charged fragments are projected onto the time- and position-sensitive detectors situated on both sides of the spectrometer. From TOF and hit positions on the detectors, the corresponding momentum vectors of the particles are reconstructed. The error bars on the experimental data points represents the statistical uncertainty.

## Computational methods

The thiophene dimer geometries required for the electronic structure calculations are optimized at the MP2/aug-cc-pVDZ level of theory using Gaussian16 program package[53]. The preferable orientations are chosen based on previously done studies[31,32]. The single (IP) and double ionization (DIP) spectra are obtained using ADC(3) approximation scheme for the one-particle Green's function[54] and ADC(2) scheme for the particle-particle propagator[55,56], respectively, in combination with cc-pVDZ basis set. The eigenstates of the ADC(n) Hamiltonian matrices are determined using the Block-Lanczos method[57,58], which is particularly useful when the breakdown of the molecular-orbital picture of ionization arises[59]. The states of the DIP spectra are classified as localized, having two holes on one of the molecules, or delocalized, having two holes distributed over two molecules in the dimer. The assignment is done within the two-hole population analysis procedure associated with the DIP-ADC(2) method and described in[60].

To better compare with experiment and minimize the intrinsic errors of the calculations, we introduced the following energy shifts of the computed ionic states (for a detailed justification of the chosen values, see the SM, section II). An energy blue shift of 1 eV is applied to the calculated DIPs[41]. Similarly, an estimated red shift of 1.2 eV is applied to the ADC(3) results in the inner-valence region above 20 eV. This value is larger than the mean error of the ADC(3) method[61] and is justified by the predominance of the 2h-1p ("two hole-one particle") states found at these energies[20], which is usual for the inner-valence region. Within the framework of the IP-ADC(3) method, the 2h-1p states are computed at first order of perturbation theory, which leads to an increase of the calculation error.

The secondary electron energy distribution is constructed from the energies of the electrons emitted via all energetically open transitions, assuming that the emitted electron energy is equal to the energy difference between the initial (IP) and final (DIP) states. The overall procedure is described in[62] and[63]. The IP spectra are Gaussian convolved with a full-width half-maximum (FWHM) of 0.4 eV to account for the vibrational broadening. The secondary electron spectra are convolved with an FWHM of 0.8 eV to account for the experimental resolution.

## Data availability

The data that support the findings of this study are available from the corresponding author upon reasonable request. The source data for Figs. 2–5 are provided with this paper as Supplementary Data 1–4, respectively. The atomic coordinates of heterocyclic dimers used for spectra calculations are provided as Supplementary Data 5.

## Code availability

This manuscript does not report original code.

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

## Acknowledgements
This work is funded by the European Partnership on Metrology, co-financed by the European Union's Horizon Europe Research and Innovation Program and by the Participating States under grant no. 21GRD02 BIOSPHERE. X.R. is grateful for support from the National Natural Science Foundation of China under Grants No. 12325406, No. 92261201 and No. 11974272; and the Shaanxi Province Natural Science Fundamental Research Project under Grant No. 2023JC-XJ-03.

## Author contributions

D.M.M., A.D.S., and A.D. conceived the project. D.M.M. and M.D. performed the experiments. D.M.M. analyzed the data. X.R. performed the AIMD simulations. A.D.S. performed the ADC calculations. D.M.M., A.D.S., X.R., and A.D. prepared the manuscript. All authors including A.I.K., A.B.T., L.S.C., and T.P. interpreted the data and commented on the manuscript.

## Funding

## Competing interests

The authors declare no competing interests.
