## [Peer review file · Communications Chemistry]

Enhanced intermolecular Coulombic decay due to sulfur heteroatoms in thiophene dimer

Corresponding Author: Dr Deepthy Maria Mootheril

Version 0:

Reviewer comments:

Reviewer #1

(Remarks to the Author)

Intermolecular Coulombic Decay (ICD) in molecular complexes is an interesting topic and useful to many readers. The approach used in this research is electron impact ionization of thiophene dimer, analysis based on Coulomb explosion, and supported by ADC(2)/cc-pVDZ level of theory.

The hypothesis of ICD is based on the observed "island" in the ion-ion coincidence plot. I see interesting theoretical results.

I have many concerns about the approach and mechanism. I do not find much uniqueness/novelty in the draft to consider it in Communications Chemistry. My observations are below.

#Coulomb explosion (CE) of ion does not remember the path of how the ion was created. Before CE, the ion can be formed by multiple pathways. The authors stated that "The events arising from the ICD process in dimers leading to (C₄H₄S⁺, C₄H₄S⁺) are separated from the background imposing the condition of small sum momentum (<20 a.u.) of the detected ions in addition to a non-zero KER", here authors reached to the conclusion that the island is due to ICD. The observed island (C₄H₄S⁺, C₄H₄S⁺) in the coincidence map is not surprising. This can be observed in other molecule dimers. Authors must check with another molecular dimer (without Sulfur-containing molecules).

In Figure 4b, the energy range of the ICD and scatter electrons are the same. This indicates that the approach used here (in caption b) is an incorrect/indirect approach, which can give an improper estimation of ICD electrons.

#There is not enough evidence/results for Enhanced Ultrafast Intermolecular Coulombic Decay. For example, the "enhanced ICD, a comparison is based on other published work [page xiv], and the "ultrafast" word is used in the title and a few times in the calculation section only without much supportive results/discussions.

Reviewer #2

(Remarks to the Author)

See attached file

Reviewer #3

(Remarks to the Author)

This article involves a joint experimental and theoretical investigation of the electron relaxation processes and subsequent Coulomb explosion events in the thiophene dimer. The goal of the paper is to prove that intermolecular Coulombic decay is a dominant pathway in the thiophene dimer and that it leads to the symmetric Coulomb explosion into two C₄H₄S⁺ monomers. Additionally, the paper concludes, using ADC(n) calculations, that ICD is especially enhanced in the thiophene dimer in comparison to other aromatic ring dimers. The enhancement is attributed to the unique overlap between the singly ionized spectrum and the region of the doubly ionized spectrum that only corresponds to a positive charge on each monomer.

Overall, the paper is clearly written and provides a nice collaboration between experimental and computational analysis. I would recommend for publication after some key modifications are addressed:

1. One general comment is that a lot of the discussion within the main text is based on figures that only appear in the SI. Additionally, the conclusions about ICD being especially enhanced in the thiophene dimer due to it being the only

energetically allowed process hinges on some of the choices of the energy shifts of the ADC(n) spectra (see point 2); this discussion also only appears in the SI. Therefore, it seems that the manuscript is better suited for a full article in comparison to a shorter communication. This would allow for a richer discussion and all the important figures to be included in the main text.

2. A major conclusion of the manuscript is that ICD is the only significant decay mechanism that can lead to the detected $2C_4H_4S^+$ Coulomb explosion event. This conclusion is mainly drawn from the electronic structure calculations (Fig. 5) that show an overlap between the single ionization spectra for the probed inner-valence ionization event and the set of doubly ionized states that only correspond to states with a positive charge on each monomer. However, the overlap with only the delocalized states will be sensitive to the choice of the energy used to shift the singly and doubly ionized spectra. The authors do a good job of explaining how they came to the choice of these energy shifts in the SI. However, it would be useful to have an expanded discussion as to how sensitive the conclusions are with respect to these choices. For example, it seems like the DIP spectrum could be blue-shifted anywhere from 0.7 eV to 1.8 eV by simply comparing the energies of the 1A1 and 1B1 orbitals at different basis sets to the corresponding experimental values. One possible suggestion, outside of expanding the text, would be to provide a range on the cyan rectangle that incorporates a lower and upper bound of the overlap between the important regions of the spectrum based on the choices.

3. Reference 34 (by many of the same authors) already performs an in-depth investigation into the Coulomb explosion event within the thiophene dimers. The focus of that work seems to be more on determining the initial structure of the thiophene dimer than on concluding unambiguously that ICD occurs. However, given that that work exists and that there is an essentially analogous plot and analysis in that work to Fig 3 in this work, it is necessary to provide a more in-depth discussion of reference 34 in the introduction and throughout the manuscript that distinguishes the novelty of this work. This is especially true for Fig 3.

4. The phrases localized and delocalized states are unnecessarily ambiguous until they are defined as states with both charges on one of the molecules or one charge on each molecule. This definition needs to come out much sooner in the main text.

5. The use of two shades of blue in the top panel of Fig. 5 leads to some confusion in that at first pass I did not know which one was being described as sky blue versus cyan. I would simply choose two different colors that can more easily be defined in the main text.

6. Was any energy shift applied to the double ionization spectra of the other dimers in section III of the SI? It is unclear based on the written text and I'm assuming the conclusions would similarly be sensitive to this choice.

Version 1:

Reviewer comments:

Reviewer #1

(Remarks to the Author)

The authors have appropriately addressed the raised points. The revised draft is improved and can be considered for publication in Communications Chemistry.

Reviewer #2

(Remarks to the Author)

The authors considered carefully all the observations of the three reviewers and modified accordingly the manuscript. The revised version is suitable for handling in Communication Chemistry

Note:

pag.VI line 111 Fig.2(c)

Reviewer #3

(Remarks to the Author)

The authors have adequately addressed my comments and concerns. I feel the paper is now ready for publication.

Reviewer #1 (Remarks to the Author):

Main comment:

Intermolecular Coulombic Decay (ICD) in molecular complexes is an interesting topic and useful to many readers. The approach used in this research is electron impact ionization of thiophene dimer, analysis based on Coulomb explosion, and supported by ADC(2)/cc-pVDZ level of theory.

The hypothesis of ICD is based on the observed "island" in the ion-ion coincidence plot. I see interesting theoretical results.

I have many concerns about the approach and mechanism. I do not find much uniqueness/novelty in the draft to consider it in Communications Chemistry. My observations are below.

Our reply: We appreciate the Reviewer for the positive evaluations of our work. Below are our point-by-point responses to the reviewer's comments and suggestions.

Comment 1: Coulomb explosion (CE) of ion does not remember the path of how the ion was created. Before CE, the ion can be formed by multiple pathways. The authors stated that "The events arising from the ICD process in dimers leading to (C₄H₄S⁺, C₄H₄S⁺) are separated from the background imposing the condition of small sum momentum (<20 a.u.) of the detected ions in addition to a non-zero KER", here authors reached to the conclusion that the island is due to ICD. The observed island (C₄H₄S⁺, C₄H₄S⁺) in the coincidence map is not surprising. This can be observed in other molecule dimers. Authors must check with another molecular dimer (without Sulfur-containing molecules).

Our reply: We correct the above-mentioned sentence in page vii of the revised manuscript to "The events arising from the CE of dimers leading to (C₄H₄S⁺, C₄H₄S⁺) are separated from the background imposing the condition of small sum momentum (<20 a.u.) of the detected ions in addition to a non-zero KER."

The conclusion on the ICD in thiophene dimers is not made from the appearance of the island in the momentum sum vs KER plot alone. The selection of the island is to separate the events from Coulomb explosion (CE) from the background in the time coincidence plot. The conclusion that part of these CE events (~86.6%) are due to the ICD process is made from the onset of the energy plot, which is below the double ionization threshold of the monomer. Similar CE is also observed in other aromatic dimers, e.g. benzene, pyridine etc. However, the appearance of the onset at ~20.6 eV well below the monomer double ionization allows us to unambiguously attribute the CE events to the ICD process. This is due to the fact that the other possible channels may not lead to charges on both monomer units of the dimer in this energy range. However, as declared on page XVII, second paragraph and in the supplementary material (IV) ~13.4% of the CE events might be arising from the sequential ionization of two monomer units

of the dimer.

Comment 2: In Figure 4b, the energy range of the ICD and scatter electrons are the same. This indicates that the approach used here (in caption b) is an incorrect/indirect approach, which can give an improper estimation of ICD electrons.

Our reply: This comment refers to Fig.3 b,c in the revised manuscript. It is an important aspect of this work that one can distinguish between the scattered projectiles and the electrons ejected from the molecules by their respective characteristic energy ranges. The scattered projectiles are predominantly more energetic. For the present 68 eV projectile energy and considering collisional energy losses of the projectile between 20 and 30 eV they have energies between 48 eV and 38 eV. In contrast the ejected electrons have predominantly low energies.

Fig.3 b shows the low energy end of the measured electron energy spectrum. In contrast in panel a) of Fig.3 data derived from scattered projectile energies, i.e. the high energy end of the electron spectrum is shown. Here one must be careful since in panel a) the energy scale does not show the scattered electron energy. We have chosen a derived energy scale (the projectile energy loss) which shows the energy difference of the detected electron with respect to the incoming projectile energy (of $E_0 = 68$ eV). Therefore, the energy loss of 28 eV in panel a) corresponds to an energy of the detected scattered electron of $68 \text{ eV} - 28 \text{ eV} = 40 \text{ eV}$.

Comment 3: There is not enough evidence/results for Enhanced Ultrafast Intermolecular Coulombic Decay. For example, the "enhanced ICD, a comparison is based on other published work [page xiv], and the "ultrafast" word is used in the title and a few times in the calculation section only without much supportive results/discussions.

Our reply: The enhancement of the low energy electron intensity due to ICD shown in Fig.3b can be quantified. If we take the peak magnitude of the spectrum with red full circles relative to the monomer spectrum with black full circles we obtain 78% enhancement. This is significantly larger than all previously observed values for other organic dimers. In the revised manuscript we included a quantitative comparison with enhancements in published works. For benzene dimers: 46% [14], tetrahydrofuran water dimers: 22% [13], pyridine dimers: 27 % [16]. Therefore, the 78 % is a significant and unprecedented enhancement. This is mentioned in Section III, second paragraph (page xiv, second paragraph).

"Ultrafast" is commonly attributed for ICD since it outperforms other relaxation mechanisms like fluorescence decay and internal conversion. As mentioned in the manuscript it is slower than Auger decay. We do not want to claim that these are a new finding of the present work. Also, the ICD transition rates are not directly determined here. Thus, we remove the word "ultrafast" from the title.

Reviewer #2 (Remarks to the Author):

Main comment:

The paper reports the clear evidence of ICD relaxation process in the electron impact ionization of thiophene dimers. The experimental results prove the role of sulfur containing inner valence bands and theoretical calculations indicate that the majority of the thiophene dimers involved has a T shape. These results have been achieved only thanks to a state-of-the-art multi-coincidence set-up and an accurate analysis of the many-body coincidence maps. The observed enhancement of the low energy ICD electrons and attributed to the presence of the sulfur atom can be of relevance in the field of radiation damage.

The paper is clearly written with a detailed reference to the literature. The supplementary material provides useful information and supports the main text. The new results, their discussion and the solid theoretical support make this work suitable for publication in Communication Chemistry.

Before the publication however the authors should consider the following points.

Our reply: We appreciate the Reviewer for the positive evaluations of our work. Below are our point-by-point responses to the reviewer's comments and suggestions.

Comment 1: Fig.4: The onset (appearance energy) of a fragment is the minimum energy to be transferred to the molecule to produce that fragment. The link of the onset to the removal of an electron from a well-defined orbital is not straightforward. Transition states may play a role and determine the value of the onset. Is this not the case in thiophene dimers?

Our reply: We have tried to address this point partly in the discussion of the projectile energy loss spectrum in Fig.3 (Fig.4 in the previous version) on page ix, first paragraph, last 9 lines. As the Referee states correctly, the appearance energy is to be transferred to the molecule to produce a particular fragment. As we understand during the very short collision time the nuclei can be considered frozen and a particular singly ionized electronic state is populated. Additionally, there is a Franck-Condon transition of the initial vibrational states to vibrational states of the ion. In principle the following detailed route from this initially populated state to the dissociation products is not relevant for our conclusions. In brief normally in such multi-atomic molecules the initial electronic excitation energy is partly or fully converted into vibrational energy via conical intersections of the potential energy surfaces of different electronic states. On this pathway barriers and transition states can be crossed finally leading to the observed fragments. Still the appearance energy allows to conclude on the initially populated singly ionized state with the restriction that the Franck-Condon overlap might not be maximum at threshold energy of a electronic state but for higher transition energy.

Comment 2: In the supplementary material a clear and convincing description of the filtering of the events due Coulomb explosion from the background is given. I would transfer some of this material in the main text instead of the single sentence on the condition of small sum momentum.

Our reply: We have transferred the sum momentum vs kinetic energy release (KER) plot and some text about the filtering of events in supplementary material I to the main text.

Supplementary material III of the previous version is moved to the main text in section III in page xiv paragraph 2. Additionally, paragraph 2 and 3 in the previous version of the manuscript are rewritten to paragraph 2 in page xiv in the revised version.

Comment 3: In the data analysis the contribution of larger clusters is mentioned. What is the expected ratio of these larger clusters with respect to the dimers, which have been estimated to be 5% with respect to the monomers?

Our reply: Measurement is done using a pulsed electron beam of 40 kHz repetition rate (25 μ s separation between two pulses). Within the accessible time of flight window of 25 μ s, clusters until thiophene trimers with mass 252 amu are detected.

A new test measurement with better mass resolution and same experimental parameters shows a better accurate estimation of 2% of dimers and \sim 0.14% trimers with respect to monomers. These numbers are corrected in the experimental methods section, line 12 in page xviii.

Comment 4: How the factor (0.3) chosen to achieve the best fit in the high energy tail may affect the comparison between theoretical and experimental electron spectra?

Our reply: The measured electron intensity is not absolute and therefore it was scaled with one factor to the theoretical spectrum. The intensity of the theoretical ICD spectrum from states with competing Auger channel was scaled down by the factor (0.3). A different scaling factor would change the relative intensities of the blue and cyan contributions in (Fig.4 in the revised manuscript top panel) and deteriorate the agreement with experiment. See the figure with the scaling factor 0.15 below.

To make the manuscript clearer in this respect we have included a short sentence from line 14 on page xiii. "This factor determines the relative contributions of the blue and cyan contributions in the top panel of Fig.4 and is chosen to provide the best visual fit between experimental and theoretical results along the high-energy tail."

Comment 5: If the spectrometer energy resolution for scattered electrons is 4 eV, why in the theoretical methods smaller broadening are considered?

Our reply: As stated in the manuscript (page xix) the theoretical binding energy spectra in Fig.4 (bottom panel) are convoluted with a FWHM of 0.4 eV to account for vibrational broadening. This is helpful to see the different bands of states and to assess their intensities while this is difficult from the multitude of individual lines before the convolution. This value for the theoretical IP spectrum is also chosen in accordance with the experimental resolution of the thiophene IP spectrum provided in supplementary material-II.1 to ensure the evaluation of the energy shift in the inner valence region. This is not related to the experimental resolution which is also different for the fast electron (about 4 eV) and the low energy electrons (0.8 eV). The effect of the spectrometer resolution to an energy loss spectrum for ionization of a defined bound state can be nicely seen in the helium data in Fig.3 (a).

Comment 5: Supplementary material II.2 : If "the second order ADC(2) for double ionization is known to have red shifts compared to experiment" why here the DIP spectrum has to be blue shifted?

Our reply: If theoretical results are red-shifted relative to the experiment - we would blue-shift the theory to better fit the experiment.

Minor points:

Abstract: ~~exceptional~~

Our reply: We have removed the word "exceptional"

Pg. iii line 3 from bottom "these principles experiments"?

Our reply: We have added punctuation marks and sentence has been modified to “Based on these principles, experiments have shown that the benzene..”

Pg.iv line 5 from bottom : “..thiophene to enhancement in “

Our reply: The sentence has been rewritten with proper punctuations for better clarity.

“Such increased non-covalent interactions can modify the valence-orbital spectrum and, in case of thiophene, lead to an enhancement in the non-local decay mechanisms, such as ICD, upon ionization from the inner-valence vacancies.”

Figure 1: Labelling the electrons would help the reader

Our reply: Electrons are labelled accordingly.

Figure 2: Color scale with number of coincidence counts would be useful

Our reply: Colormap is changed and counts for the color scale are added.

Pg. ix line 2 from bottom: “bottom”

Our reply: We have changed the figure numbers to fig. 3 a, b and c. Correspondingly, the captions are changed.

Pg.xi Define DIP and IP in the main text and not in the caption of Figure 5

Our reply: We have moved the definitions from caption of figure 4 in the revised manuscript to line 1 paragraph 2 of page ix and line 2, paragraph 3 of page xi.

Reviewer #3 (Remarks to the Author):

Main comment:

This article involves a joint experimental and theoretical investigation of the electron relaxation processes and subsequent Coulomb explosion events in the thiophene dimer. The goal of the paper is to prove that intermolecular Coulombic decay is a dominant pathway in the thiophene dimer and that it leads to the symmetric Coulomb explosion into two $C_4H_4S^+$ monomers. Additionally, the paper concludes, using ADC(n) calculations, that ICD is especially enhanced in the thiophene dimer in comparison to other aromatic ring dimers. The enhancement is attributed to the unique overlap between the singly ionized spectrum and the region of the doubly ionized spectrum that only corresponds to a positive charge on each monomer.

Overall, the paper is clearly written and provides a nice collaboration between experimental and computational analysis. I would recommend for publication after some key modifications are addressed:

Comment 1: One general comment is that a lot of the discussion within the main text is based on figures that only appear in the SI. Additionally, the conclusions about ICD being especially enhanced in the thiophene dimer due to it being the only energetically allowed process hinges on some of the choices of the energy shifts of the ADC(n) spectra (see point 2); this discussion also only appears in the SI. Therefore, it seems that the manuscript is better suited for a full article in comparison to a shorter communication. This would allow for a richer discussion and all the important figures to be included in the main text.

Our reply: The sum momentum vs kinetic energy release (KER) plot and some text about the filtering of events are transferred to the main text in page vi paragraph 4.

Section III of supplementary and the figure 5 of supplementary is transferred to the main text to section III in page xiv paragraph 2. The total words of the text is checked to be within the limit.

Comment 2: A major conclusion of the manuscript is that ICD is the only significant decay mechanism that can lead to the detected $2C_4H_4S^+$ Coulomb explosion event. This conclusion is mainly drawn from the electronic structure calculations (Fig. 5) that show an overlap between the single ionization spectra for the probed inner-valence ionization event and the set of doubly ionized states that only correspond to states with a positive charge on each monomer. However, the overlap with only the delocalized states will be sensitive to the choice of the energy used to shift the singly and doubly ionized spectra. The authors do a good job of explaining how they came to the choice of these energy shifts in the SI. However, it would be useful to have an expanded discussion as to how sensitive the conclusions are with respect to these choices. For example, it seems like the DIP spectrum could be blue-shifted anywhere from 0.7 eV to 1.8 eV by simply comparing the energies of the 1A1 and 1B1 orbitals at different basis sets to the corresponding experimental values. One possible suggestion, outside of expanding the text, would be to provide a range on the cyan rectangle that incorporates a lower and upper bound of the overlap between the important regions of the spectrum based on the choices.

Our reply: An expanded discussion on the effect of the DIP blue shift on the overlap between inner-valence orbitals and the energy gap between the ICD and Auger thresholds is provided in Supplementary Material Section II.2. Additionally, theoretical double and single ionization spectra with different DIP shifts (0.7 eV, 1.0 eV, and 1.8 eV) are included to illustrate the overlap region. The figures demonstrate that this overlap remains largely unchanged regardless of the chosen DIP shift. Therefore, our conclusion regarding the enhancement of low-energy electrons remains valid, irrespective of the selected DIP shift.

Comment 3: Reference 34 (by many of the same authors) already performs an in-depth investigation into the Coulomb explosion event within the thiophene dimers. The focus of that work seems to be more on determining the initial structure of the thiophene dimer than on concluding unambiguously that ICD occurs. However, given that that work exists and that there is an essentially analogous plot and analysis in that work to Fig 3 in this work, it is necessary to provide a more in-depth discussion of reference 34 in the introduction and throughout the manuscript that distinguishes the novelty of this work. This is especially true for Fig 3.

Our reply: The following lines are added to the introduction page iv

‘A similar coincidence measurement on the thiophene dimer exists, detecting ion pairs undergoing CE and conducting an in-depth investigation of the dimer’s structure in a supersonic gas jet [33]. This study also determines the potential energy curve (PEC) of thiophene dimers in different conformations using the measured kinetic energy release spectrum of the ion pairs. However, coincident electron detection was not performed, which is crucial for confirming the ICD process and measuring ICD electron energy.’

and page viii paragraph 2 line 2,

‘The KER spectrum is analogous to Fig. 4 in [33].’

Comment 4: The phrases localized and delocalized states are unnecessarily ambiguous until they are defined as states with both charges on one of the molecules or one charge on each molecule. This definition needs to come out much sooner in the main text.

Our reply: The definition for ‘delocalized’ and ‘localized’ states is added in line 7 and 8 of paragraph 3 of page xi.

Comment 5: The use of two shades of blue in the top panel of Fig. 5 leads to some confusion in that at first pass I did not know which one was being described as sky blue versus cyan. I would simply choose two different colors that can more easily be defined in the main text.

Our reply: The color of the direct decay part of the ICD electron spectrum is changed to green in the Fig. 4 (top panel) of the revised manuscript. The color of the competing part is unchanged. Corresponding changes are made in the text (page xi, line 2 from the bottom and page xiii, line 3 from the top and lines 8,9 from the bottom) and the figure caption.

Comment 6: Was any energy shift applied to the double ionization spectra of the other dimers in section III of the SI? It is unclear based on the written text and I’m assuming the conclusions would similarly be sensitive to this choice.

Our reply: Same energy blue shift of 1 eV is applied to the double ionization thresholds of other dimers as well (figure 5 in the revised manuscript) and mentioned in the main manuscript. The following sentence is added to the second paragraph of discussion section page xiv,

“Similarly, an energy blue shift of 1 eV is applied to the indicated double ionization thresholds of both dimers and monomers.”

The paper reports the clear evidence of ICD relaxation process in the electron impact ionization of thiophene dimers. The experimental results prove the role of sulfur containing inner valence bands and theoretical calculations indicate that the majority of the thiophene dimers involved has a T shape.

These results have been achieved only thanks to a state-of-the-art multi-coincidence set-up and an accurate analysis of the many-body coincidence maps.

The observed enhancement of the low energy ICD electrons and attributed to the presence of the sulfur atom can be of relevance in the field of radiation damage.

The paper is clearly written with a detailed reference to the literature. The supplementary material provides useful information and supports the main text. The new results, their discussion and the solid theoretical support make this work suitable for publication in Communication Chemistry. Before the publication however the authors should consider the following points.

- a) Fig.4: The onset (appearance energy) of a fragment is the minimum energy to be transferred to the molecule to produce that fragment. The link of the onset to the removal of an electron from a well defined orbital is not straightforward. Transition states may play a role and determine the value of the onset. Is this not the case in thiophene dimers?
- b) In the supplementary material a clear and convincing description of the filtering of the events due Coulomb explosion from the background is given. I would transfer some of this material in the main text instead of the single sentence on the condition of small sum momentum.
- c) In the data analysis the contribution of larger clusters is mentioned. What is the expected ratio of these larger clusters with respect to the dimers, which have been estimated to be 5% with respect to the monomers?
- d) How the factor (0.3) chosen to achieve the best fit in the high energy tail may affect the comparison between theoretical and experimental electron spectra?
- e) If the spectrometer energy resolution for scattered electrons is 4 eV, why in the theoretical methods smaller broadening are considered?
- f) Supplementary material II.2 : If "the second order ADC(2) for double ionization is known to have red shifts compared to experiment" why here the DIP spectrum has to be blue shifted?

Minor points

Abstract: ~~exceptional~~

Pg. iii line 3 from bottom "these principles experiments" ?

Pg.iv line 5 from bottom : "...thiophene to enhancement in "

Figure 1: Labelling the electrons would help the reader

Figure 2: Color scale with number of coincidence counts would be useful

Pg. ix line 2 from bottom: "bottom"

Pg.xi Define DIP and IP in the main text and not in the caption of Figure 5